# Working with Goals and Trauma in Youth Mental Health

**DOI:** 10.3390/ijerph191711048

**Published:** 2022-09-03

**Authors:** Duncan Law

**Affiliations:** Faculty of Brain Sciences, Division of Psychology and Language Science, University College London, London WC1E 6BT, UK; ucjtdj1@ucl.ac.uk

**Keywords:** trauma, goals, GBO, goal-oriented, ROMs, I-PROMs

## Abstract

There is good evidence of the value of working with goals in youth mental health services and settings. As such, goal-oriented practice is seen as a core component of good mental health interventions. Yet, there is debate among clinicians and academics about whether working with goals is a valid approach with clients who have experienced trauma. In this paper, I will explore the impacts of trauma and argue that working with goals, including the use of tools that facilitate these practices, such as the Goal-Based Outcome (GBO) tool, is as vital when working in trauma-informed interventions, as with any other mode of practice or client group.

## 1. Trauma and Mental Health

Shared decision making (SDM) is a central tenet of health policy [1]. Collaborative goal setting is seen as a vital part of SDM [2]. In youth mental health services, goal setting helps young people feel more included in their care [3,4] and facilitates better therapeutic alliance [5]. As such, goal-oriented practice is seen as a core component across most therapeutic modalities [6] and has demonstrated utility across a range of presenting issues [7,8]. Yet, there is debate among clinicians and academics about whether working with goals is a valid approach with clients who have experienced trauma, and who might benefit from longer term mental health interventions [9,10]. Goal setting and the use of tools to assist in goal-oriented practice such as the Goal-Based Outcome (GBO) tool appear to be seen by some as too simplistic to be used with youth who present with complex histories. Even with emerging evidence of the value of feedback and outcome tools in clinical practice [11], there appears to be a reluctance to use these tools with youth who have experienced trauma [10]. To understand this reticence, we need to take a step back to understand the impact of trauma and mental health.

There is compelling evidence of the links between trauma and mental illness [12,13]. Trauma, and especially complex trauma [14], seems to have a tripartite of negative effects on mental health and well-being:First, people who experience trauma, particularly in childhood, are much more likely to develop mental health issues [15];Second, if they do develop a mental illness, they are less likely to engage in help-seeking behaviour for their mental distress [16,17,18];Third, even if they do get to the point of being face-to-face with a mental health professional, outcomes of therapy tend to be less effective [19].

McCrory and colleagues describe the effects of trauma on the developing brain [20], leading to what they describe as ‘latent vulnerabilities’ to developing mental health issues in later life. People who experience trauma learn to perceive others as less trustworthy [21]. Consequently, they are likely to interact with the world in a way that is likely to lead to social isolation. This ‘Social thinning’ in turn exacerbates the impact of the trauma [22]. In essence, the experience of trauma, and the person’s attempts to adapt and protect themselves from further trauma, tip them into a trajectory that negatively affects their interactions and experiences of the world and people in it, and interferes with their ability to learn and adapt, increasing the likelihood of mental health issues [20].

The field of natural pedagogy offers insight into how children learn to manage the complexities of the human world [23]. They suggest that we do not learn from people randomly, but rather learn from others whom we feel are a trusted source of beneficial information. Evidence suggests that children are more likely to learn from people who show interest in them [24]. Communicators that show particular interest in the listener engender trust and cue the listener that the information about to be imparted is of significance and beneficial to them and they should listen closely to the information that is about to be imparted. These so called ‘ostensive cues’ are triggered when “*the communicator explicitly recognizes the listener as a person with intentionality*” p. 589 [25]. Ostensive cues to attend seem to help children to open their minds to take on board cultural knowledge and generalize learning across settings, helping them to adapt and function well in complex human societies [26].

Fonagy and colleagues argue that these ostensive cues are the roots to children having ‘epistemic trust’ when presented with a source of learning. Epistemic trust engenders a state of openness to learning and adaptation. They propose that ‘Mentalization’ is the mechanism that underlies the development of epistemic trust [26]. Mentalization is: “*a form of imaginative mental activity …. perceiving and interpreting human behaviour in terms of intentional mental states (e.g., needs, desires, feelings, beliefs, goals, purposes and reasons)*.” p. 47 [27]. Put simply, it is the experience of another person taking interest in our mind, and in our intentional mental states, which builds understanding and trust between people. This experience of being mentalized, is the mechanism that facilitates epistemic trust between the receiver of human cultural knowledge and the trusted source of learning [26].

In summary, the accumulation and generalization of learning, which is necessary to adapt and understand the complexities of human society, needs to come from a source which the learner views as trustworthy: a state more likely to occur when a child experiences a person who shows interest in them, their minds, and their intentions, needs, desires, purposes, or in other words, their ‘***goals***’.

Trauma negatively impacts on a person’s ability to be open to learning and adaptation. Trauma, particularly at the hands of a previously trusted source of learning (a caregiver), leads to a breach of epistemic trust. If the trauma is significantly impactful, this can lead to a state of ‘epistemic hypervigilance’ or ‘mistrust’ where people learn that previously trusted sources of learning are no longer reliable, and this can generalize out to all other sources of learning [26]. In other words, the child adopts a state that doubts all sources of information and effectively learns that it is unsafe to learn from others [28].

Psychological therapy arguably works only if people are open to learning new ways of thinking, understanding, and behaving [29,30], creating new meanings of their experiences in order to understand and interact with the world differently [31]. Therapists and mental health practitioners who experience people in this state of epistemic hypervigilance sometimes describe it as being faced with a ‘brick wall’ between them and the person they are trying to help [32]. The therapist’s task is to establish a trusted space where the client can begin to re-establish trust in human relationships and re-adapt from a learnt state of epistemic hypervigilance to one of epistemic trust with the therapist. The establishment of trust in a therapeutic setting slowly dismantles the ‘brick wall’ of epistemic hypervigilance, one brick at a time, through the experience of connection with a benign helper. This can begin to establish a sense of trust through an experience of someone showing intention to care. The experience of being mentalized begins to re-establish an experience of being cared for and therefore begins to develop trust and the possibility of re-adapting to a stage where learning is once again possible [26].

## 2. Working with Goals and Working with Trauma

So, what has this got to do with working with goals and goal-oriented practice? Well, if we accept the clinical implications from the previous section on trauma and mental health, then the therapist’s task becomes clear: the therapist must work to create a therapeutic space where the client feels safe enough to share something of what they want to be different in their lives: their goals.

This is not an easy task in my clinical experience, and clinicians who work with trauma often describe how difficult it is to have a meaningful conversation about goals with traumatized clients. Of course, this will be the case if we accept the roots of trauma and the mechanisms that underlie it. The client who is in a state of epistemic hypervigilance will have learnt not to risk telling people what they want. The youth has learnt that help is not available; admitting that things are not how they might want them to be is to show some aspect of vulnerability and the fear is that this vulnerability will be used against them. The therapist’s experience of the youth is of someone with whom they find it hard to have conversations about goals. My sense is that this experience is overly interpreted by clinicians as the person ‘not knowing what they want’, whereas I would argue that this is more likely to be someone ‘not daring to share what they want’ and not yet trusting us enough to risk showing such vulnerability.

Clues as to how we create this space come from the field of natural pedagogy discussed earlier [23]: the teacher/therapist needs to signal the intention to impart important knowledge. These ‘ostensive cues’ orient the learner/client, that there may be something helpful to be gained from the teacher/therapist. Mentalization describes the therapeutic mechanism that re-establishes trust as the client’s experiences being understood as a person with a mind of their own and who has internal desires, wants, intentions, and goals [26]. This begins to break down the client’s adaptive, epistemic mistrust, opening up new channels of communications and facilitating adaptation and learning. This signals intention to teach and intention to care, building the foundations of effective therapy.

This need to signal intention-to-care is backed up in the work of Kate Martin, who demonstrates that young people who have experienced trauma can be cued by therapists and care workers, that they are being invited to participate in decisions about their care. These cues signal to the young person that the care-worker is inviting them to be more fully involved in their care and orientates them to be more active and engaged. In turn, this engenders a sense of inclusion and involvement that, without these cues, might be opportunities that were otherwise missed [33]. These cues demonstrate intention to care, build trust and involvement, and help create space where learning and adaptation are more likely.

Seeking to uncover and understand a young person’s wishes, desires, hopes, and intentions—or in short, their goals—are fundamental to both mentalization and SDM. The GBO tool, is a client-defined feedback tool [34] or ideographic patient-reported outcome measure (I-PROM) [35] developed to help facilitate collaborative goal-oriented conversations, set goals with the client, and to help track progress towards goals. In turn, this can facilitate further collaborative goal-oriented conversations [34]. Youth are invited to set, usually, up to three goals about what they want to be different in their lives as a consequence of engaging in therapy, and rate progress toward these goals on a simple, 11-point scale. The tool can be re-rated as appropriate, throughout the intervention to track progress (or lack of progress) [36]. Used well, as an integrated part of a sound therapeutic processes, the GBO is a tool that facilitates and signals interest in what the young person wants to be different in their lives. Furthermore, it signals intention to care: ‘*I want to know what you want, as I recognize you as an individual with intentions, wishes and hopes, and I want to understand what you want in order to help you to be able to get there*’.

Let me be absolutely clear at this point: I am not arguing that merely asking about goals, i.e., what someone wants to be different in their life, is sufficient to create this necessary shift from epistemic hypervigilance to a state of openness to learning. The therapeutic and human processes that lead to the shift are multiple and complex. However, I am stating that part of this complex process, an important part of this complex process, is to show interest in what somebody wants to be different in their life, i.e., their goals. Additionally, I would go as far as to say that if we do not show interest in what people want to be different in their lives (their goals) then we risk the possibility of reducing the impact of establishing therapeutic trust.

My sense is that one of the reasons therapists are reluctant to use goal setting in trauma-informed practice is that often the use of goal-based outcomes has become a bureaucratic *task*: to set a goal and monitor progress—only for the data to supply some distant, centralized system. This undermines the more important aspect of working with goals, which is the *process* of using tools, such as the GBO, to engage with a person to understand their story, their background, and their future-oriented intentions and wishes in order to create this atmosphere where the client experiences the therapist’s intention to care. Using tools such as the GBO is a signal of interest in the client’s intentional states, where the client has an experience of being mentalized by the therapist, which opens a possibility of building trust, breaking down the wall of mistrust, and allowing a space where adaptation and learning can begin. I would argue strongly that this *process* of goal-oriented practice is the primary purpose that underpins the value of all client-defined feedback tools and I-PROMs [35] such as the GBO. The aim of the GBO tool is to remind the practitioner of the fundamental need to seek to understand the client’s hopes and wishes, and to remind them to show interest in the client’s intentional mind [34]. Once, and only once, we have laid the foundations of trust, can we and should we begin the task of goal setting: reaching agreement on the aims of the therapy, and finally begin rating the goals set.

## 3. Case Example

Youth Z was a 15-year-old who identified as female. She was known to have witnessed domestic abuse from her father to her mother and was later the victim of sexual exploitation. She was removed from home and accommodated into a secure children’s home (SCH) aged 14 and was later referred to a local child mental health service. Prior to the referral, and in consultation with a Clinical Psychologist, the staff at the SCH had spent time getting to know Z and ‘hearing her story’ to understand her behaviour in the context of her life history and the associated trauma. This work had helped lay important foundations that served as a platform for the start of therapy.

In the first session, Z was largely silent and showed behaviours that suggested she would rather not be with the therapist, but she did not make to leave the therapy room. The therapist wondered out loud why she might be quiet and made reference to it being difficult to trust new people and how this made sense if she had experienced other adults in her life as being untrustworthy. Z returned for a second session and, when invited, began to share her ‘story’ of what had happened to her in her past. She spoke of how the memories get “*stuck in my head*” and this made her feel fearful and angry. After some further discussion of past experiences and Z’s present feelings and behaviours, the therapist asked what Z might want to be different in her life and how therapy might help. At first Z said, “*I don’t know*”; the therapist made a tentative suggestion about Z’s angry and frightened feelings and wondered if these might be something she wanted to be different. The therapist reminded Z of the link she had made between the memories ‘stuck in her head’ and the strong feelings she experienced. Z spoke about how these strong feelings got her into trouble, often being aggressive and verbally abusive with new members of staff at the SCH and other young people who entered the home. The therapist empathized with her about how difficult this must be and guessed that Z might feel isolated and lonely as a result but how Z might also want to make connections with others her age and get on better with the care staff, but this might feel unsafe. Z nodded in half-agreement.

In the third session, these themes were revisited; by now, Z has experienced the therapist signalling intention-to-care, showing interest in her and her mind. The therapist had imagined what it might be like to be Z and the links between her intentional mental states and her behaviours (mentalization). When the theme of what Z wanted to be different was revisited, she was now ready to engage in a discussion about her wishes for change (goals) relating to her fear and anger from the past and the impact these had on her current behaviours, in an attempt to protect herself from further harm. Z and the therapist began to shape broad goals around “*feeling less alone*” and “*not always being in trouble*”, and Z was able to rate where she felt she was in the path towards achieving those goals, using the GBO.

The therapy that ensued focused on ‘unsticking thoughts’ or ‘trauma processing’. The goals of therapy were revisited from time to time throughout the intervention, and although Z was sometimes reluctant to rate the goals, the conversations that followed these discussions seemed to facilitate a better working relationship. Towards the end of therapy, Z was asked about her experience of goal setting in the early sessions. She recounted that she felt the therapist was showing a real interest in her as a person and not just as a “*problem*” and seemed interested in what she genuinely wanted to be different in her life, as opposed to what the authorities expected her to change. She expressed that this was one of the important experiences that had helped her to engage in the work, albeit reluctantly at times.

## 4. Conclusions

In conclusion, our increasing knowledge of the effects of trauma predicts clients who have adapted to become more cautious to share and to be less open to learning and adaptation. Showing interest and intention-to-care can begin to break-down rigidity and slowly rebuild trust. Working with goal processes, facilitated by I-PROMS such as the GBO can remind the therapist of the need to show interest in the client’s intentional mental states and give the client the experience of being with someone who wants to understand them and their mind. I would argue that rather than avoiding goal-oriented conversations with people who have experienced trauma because of the difficulty and complexity in doing so, trying to engage with youth about what they want is a necessary part of the therapeutic process; not least with people who have experienced trauma. Conversations about a young person’s future hopes and wishes (their goals,) facilitated by tools such as the GBO, are likely to enhance a person’s experience of being cared for rather than interfere with it.

We need to shift our therapeutic stance away from blaming the client for ‘*not knowing what they want*’ to one where the therapist takes more responsibility for creating a space where the client feels safe enough to ‘*dare-to-share*’ their goals and wishes. The therapist uses the same core therapy skills to facilitate a good therapeutic alliance and create an environment for therapeutic change (well documented elsewhere [37]), with the specific focus being to help the client discover, decide, and choose where they want to get to in therapy. I would argue that the need for this stance is as vital in trauma work as it is in any other therapeutic encounter. We need to do more, not less of this; of course, we need to be mindful of when we feel a client is not ready to share their hopes and dreams and desires and wishes in life, but we must, at all costs, avoid an assumption that they do not have any goals. Such views take an overly professionalized, paternalistic stance that disempowers and infantilizes, when what we should be doing is engaging and empowering.

The challenge is to try harder, involve more, signal louder, and cue better our intentions to understand and care. None of these things are easy, but if the ideas that sit behind the GBO and other I-PROMs can play a part in this complex task, we should be using them more, not less, with youth who have experienced trauma. The quality of data we get from client-defined tools will be better and the impact on therapy is likely to be greater.

## Data Availability

Not applicable.

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
