# Peer review of "Working with Goals and Trauma in Youth Mental Health"

_ijerph, 2022, doi:10.3390/ijerph191711048_

Round 1

Reviewer 1 Report

This is a well written paper but not sure if it qualifies as an original article based on its format. It is very interesting to read and makes a very good argument for goal-oriented approaches in addressing mental health care for adolescents.

The author can elaborate on the context or history in relation to the debate among clinicians and academics about this topic. I am also left with a question in my mind: what are alternatives to goal-oriented approach and what are the pros and cons of these alternatives. I think this will help the reader assess the value of the paper.

I would also recommend more description be added to the GBO tool (page 3, line 140). What does it consist of? How is it administered? etc...

In relation to the concept of 'dare-to-share', what if adolescents really do not know what they want? How would a goal-oriented approach address this? Would the goal be to assist them in doing so? What are the steps? Is this dynamic? Should we be directing our attention to a platform of potential goal(s)?

Author Response

Dear Reviewer

thank you for your very helpful comments. i have reviewed them carefully and made appropriate changes as follows (in green) :

This is a well written paper but not sure if it qualifies as an original article based on its format. It is very interesting to read and makes a very good argument for goal-oriented approaches in addressing mental health care for adolescents. i agree, it is more of a theoretical report which i was invited to write by the guest editors of this special edition of the journal but i couldn't see a way of submitting it under that heading  - hence it think it must have been sent out to reviewers as if it were a research article. It is more somewhere between an opinion piece and a theoretical report. The other corrections i have made have been in the light of this 're-classification'

The author can elaborate on the context or history in relation to the debate among clinicians and academics about this topic. - this has now been added in the first paragraph of the of the paper to help set the context more - thank you

I am also left with a question in my mind: what are alternatives to goal-oriented approach and what are the pros and cons of these alternatives. I think this will help the reader assess the value of the paper. The focus of the paper is on using the GBO with youth who have experienced trauma and making the case for goal-oriented practice. in effect the whole paper is addressing the pros and cons of this way of working. I was specifically invited to write a paper by the guest editors to this effect. I feel there isn't scope to cover alternative ways of working with trauma  - of which there is much debate in the field and would require a whole other article to address this.  i hope the references in the early section of this paper will direct readers to appropriate authors and articles who cover these issues in great depth. As we agree this isn't a regular research article, but more of a theoretical report or opinion piece. In the light of this the usual conventions of a research paper don't quite fit.  The whole article is effectively a critique and recommendations for practice.  i hope this suffices?

I would also recommend more description be added to the GBO tool (page 3, line 140). What does it consist of? How is it administered? etc... A description of the GBO and administration is now added - thank you

In relation to the concept of 'dare-to-share', what if adolescents really do not know what they want? How would a goal-oriented approach address this? Would the goal be to assist them in doing so? What are the steps? Is this dynamic? Should we be directing our attention to a platform of potential goal(s)? I have added some additional material and a case example which i hope address the important questions you raise.

Thank you once again for your very helpful review. I hope you feel the changes made are adequate in the light of the 're-classification' of the paper

All the best

D

Reviewer 2 Report

Dear Author, I've read your interesting investigation, and I believe it provides relevant information for practice and future research.

Below I provide some suggestions and comments with the final aim to help to improve the quality of this manuscript and its reporting.

Minor comments: some typos and spelling mistakes should be revised as references according to the journal's style.

General suggestions: 

- First of all, I believe this type of report fits better as commentary letter or brief theoretical report rater than article. No novel data is provided.

- In any case, I believe A case study should be added to add to the knowledge and become relevant to the scientific community.

- Ideographic patient reported outcome measures (I-PROMs) appears at the end of the manuscript and I believe more insights and related-previous literature in this field could and should be included, considering that this is one keyword in the list of the manuscript's abstract. 

- Future directions and implications for practice should be strengthen at the end of the theoretical explanation to reinforce the point and as a separate section (not just included as summary in conclusions) Also, criticisms to this approach or gaps in the scientific literature might be included and highlightened. 

Author Response

Dear reviewer

thank you for your very helpful comments. i have reviewed them carefully and made appropriate changes as follows (in green) :

Minor comments: some typos and spelling mistakes should be revised as references according to the journal's style. - Typos corrected and referencing now in journal style

General suggestions: 

- First of all, I believe this type of report fits better as commentary letter or brief theoretical report rater than article. No novel data is provided. - i agree, it is more of a theoretical report and this is was i was asked to write by the guest editors of this special edition of the journal but i couldn't see a way of submitting it under that heading  - hence it think it must have been sent out to reviewers as if it were a research article. It is more somewhere between an opinion piece and a theoretical report. The other corrections i have made have been in the light of this 're-classification'

- In any case, I believe A case study should be added to add to the knowledge and become relevant to the scientific community. - case study added  - thank you for the suggestion

- Ideographic patient reported outcome measures (I-PROMs) appears at the end of the manuscript and I believe more insights and related-previous literature in this field could and should be included, considering that this is one keyword in the list of the manuscript's abstract.  - as this article was commissioned by the guest editors for a special edition of the journal on routine outcome measures, including I-PROMS, my sense it that the readership will understand the context. The term I-PROMS is a newly coined in the Sales paper referenced, so there isn't much history to add - i hope this is acceptable?

  • Future directions and implications for practice should be strengthen at the end of the theoretical explanation to reinforce the point and as a separate section (not just included as summary in conclusions) Also, criticisms to this approach or gaps in the scientific literature might be included and highlightened. - as we agree this isn't a regular research article, but more of a theoretical report. In the light of this the usual conventions of a research paper don't quite fit.  The whole article is effectively a critique and recommendations for practice. I have added some additional material to strengthen the practice implications - i hope this suffices?

Thank you once again for your very helpful review. I hope you feel the changes made are adequate in the light of the 're-classification' of the paper

All the best

D